# The Contribution of Organised Leisure-Time Activities in Shaping Positive Community Health Practices among 13- and 15-Year-Old Adolescents: Results from the Health Behaviours in School-Aged Children Study in Italy

**DOI:** 10.3390/ijerph17186637

**Published:** 2020-09-11

**Authors:** Alberto Borraccino, Giacomo Lazzeri, Omar Kakaa, Petr Bad’ura, Daniele Bottigliengo, Paola Dalmasso, Patrizia Lemma

**Affiliations:** 1Department of Public Health and Paediatrics, University of Torino, 10124 Torino, Italy; alberto.borraccino@unito.it (A.B.); paola.dalmasso@unito.it (P.D.); patrizia.lemma@unito.it (P.L.); 2Department of Molecular and Developmental Medicine, University of Siena, 53100 Siena, Italy; giacomo.lazzeri@unisi.it; 3Institute of Active Lifestyle, Faculty of Physical Culture, Palacky University, 77147 Olomouc, Czech Republic; petr.badura@upol.cz; 4Department of Cadiac, Thoracic, Vascular Sciences and Public Health, University of Padova, 35122 Padova, Italy; daniele.bottigliengo@studenti.unipd.it

**Keywords:** organized leisure-time activities (OLTA), physical activity recommendations, health practices, alcohol and tobacco use, life satisfaction, HBSC

## Abstract

*Background and Objective*: Participation in organised out-of-school leisure-time activities (OLTAs) has been shown to have a positive impact on community health practices and to lessen inequities in social and environmental opportunities among youths. According to the social capital theory, OLTAs foster bridging ties that allow individuals to forge new, wider-ranging social connections, increasing social integration and opportunities for identity-related exploration. This study aimed to describe participation in different types of OLTAs and its association with perceived life satisfaction, physical activity, tobacco use, alcohol consumption, and drunkenness in a representative sample of youths. *Methods:* A representative sample of 11-, 13- and 15-year-old students (*n* = 47,799) was recruited throughout all Italian regions within the Italian 2013/2014 Health Behaviours in School-aged Children (HBSC) study. Data were collected according to the HBSC study protocol. Participants were compared on outcomes according to OLTA participation type (i.e., non-sport, sport-only, and mixed vs. no-OLTA). In accordance with the study sampling procedures, hierarchical logistic regression models were used. *Results:* Participation in OLTAs was significantly associated with high life satisfaction in all ages (sport-only: odds ratio (OR) = 1.67, 1.48 and 1.55 for 11- 13- and 15-year-olds; mixed: OR = 1.95, 1.60 and 1.45, respectively). Youths participating in OLTAs were more likely to meet physical activity recommendations and report lower rates of tobacco use and drunkenness. *Conclusions:* Participation in OLTAs showed a favourable impact on health behaviours. Thus, community organisations and clubs, whether supported by public investments, could contribute to the enhancement of beneficial health practices, by engaging and serving the community as a whole and further reducing inequities in both social and environmental opportunities.

## 1. Background

There is evidence that unhealthy behaviours, which range from lack of physical exercise to tobacco smoking and excessive alcohol consumption, established during adolescence tend to persist into adulthood having a detrimental effect on population health [1,2]. According to Frolich et al. unhealthy behaviours among adolescents tend to reflect collective lifestyles, rather than being mainly individual choices. These behaviours are, indeed, the result of adolescents’ continuous adaptation to the structures that exist and the opportunities that are offered within their micro context of reference, which includes both their social and physical environment [3]. Individual experiences can, in fact, be encouraged or hindered by characteristics of the physical environment, available social relationships and consequential expectations of perceived appropriate behaviours. Therefore, health behaviours have to be interpreted within a broader community perspective as health practices rather than risk/protective factors. This shift in focus from health behaviours to health practices is a prerequisite to tackling inequalities in health, as it moves the emphasis from inequalities in outcomes to inequities in opportunities [4].

According to the social capital theory developed by Putnam (2000), community members share different types of social ties at different points in their lives: bonding ties characterise socially homogeneous groups, such as families; whereas bridging ties refer to the weaker social ties developed between socially heterogeneous groups, such as choirs, sportive, or intellectual clubs [5,6]. When measured bridging and bonding social ties showed to be linked to community health improvement, in particular in shaping positive health behaviours, otherwise discussed as community health practices [4,5]. However, according to Granovetter (1983), bridging ties are more important in adolescence, as they allow individuals to widen their connections beyond those available in their proximal social circle. These wider connections can provide a greater familiarity with the community and its resources, which in turn means that youths are aware of and can access more information and services when needed [7]. Based on these results, participation in organised groups and associations showed that bridging ties can be a resource for social integration offering distinct opportunities for identity-related exploration. More recently, such goal-oriented extracurricular programmes conducted in community-based settings and supervised by adults (e.g., team sports, music, theatre, choirs, or religious groups) were gathered under the definition of *Organized* “leisure-time” *activities (OLTAs).* Under this term are gathered a broad range of adult-sponsored activities that fall outside the regular school curriculum and include diverse contexts such as school-based extracurricular activities, community organizations, and youth development programs [8,9,10].

OLTAs have been shown to act as a protective factor against several high-risk behaviours, including substance abuse and lack of physical activity, but also delinquency and bullying and have also been related to positive school engagement, improved academic achievement, higher life satisfaction, and better self-rated health [11,12,13,14]. However, this may depend on the type of OLTA, with sportive OLTAs playing a prominent role in almost all the considered outcomes, here comprising life satisfaction [11,15]. Indeed, compared with other types of OLTA, sportive OLTAs have the highest overall rate of participation, as well as ‘unique’ diverging outcomes, like the elevated risk of alcohol consumption that is sometimes reported in this group [16,17]. Moreover, the associations between participation in OLTAs and health behaviours were shown to differ by gender (involvement predicted lower levels of risky behaviours in boys) and socioeconomic status, youths from low-income families benefited more from extracurricular participation than those from high-income families, thus reinforcing the need to deepen our knowledge on the importance of bridging ties in youths participating in different types OLTAs in relation to youth health behaviours at a national level [13,14,18,19,20]. OLTA in many European countries and in Italy, as well, are offered through the organised effort of the so called third-sector. The ‘third-sector’ is an umbrella term that covers a range of different organisations with different structures and purposes, belonging neither to the public sector nor to the private sector (namely profit-making and private enterprises). Third-sector organizations are non-profit, as they raise funds and generate financial surpluses to invest in social, environmental, or cultural objectives. Such organisations are values-driven, pursuing specific goals aligned with particular social, cultural or political perspectives; they mainly operate at the very local level into a national perspective, in particular for sport associations. Access to OLTAs in Italy is partly guaranteed by public funds but mainly determined by household financial contributions [21].

Given the above, the aim of the study was to describe participation in different types of OLTAs, and to evaluate associations between this participation and perceived life satisfaction, physical activity, tobacco use, alcohol consumption, and drunkenness in a national representative sample of Italian youths, while also taking into account geographical context, gender, and socio-economic status (SES).

## 2. Data and Methods

### 2.1. Compliance with Ethical Standards

The study protocol, questionnaire and all detailed procedures were in accordance with the 1964 Helsinki Declaration and its later amendments.

Informed consent was obtained from all individual participants included in the study at local authority, school, parent/guardian, and student levels. No reimbursement was planned nor provided for participation.

The Italian version of the Health Behaviour in School-aged Children (HBSC) study protocol and whole procedures were formally approved on 21 March 2012 by the Bioethics Committee of the University of Torino, Project Identification Code—Stili di vita e Progetto Salute/03/12.

### 2.2. Study Population and Design

Data were collected as part of the 2013/14 HBSC study in Italy. The HBSC study is a World Health Organisation (WHO) collaborative cross-national survey, which runs every 4 years in more than 40 participating countries, all adhering to a detailed international study protocol [22].

School is the primary sampling unit, drawn by systematic cluster sampling of all public and private schools throughout all Italian regions. On the behalf of the Ministry of Education, a total of 3315 classes agreed to participate in the study, resulting in a representative sample of youths aged 11, 13, and 15 years, with a response rate of 90.1% and a final sample of 47,799 students. A total of 4256 students with incomplete answers in the variables under analysis were excluded. Therefore, as per the study aims, the analyses comprised a final sample of 43,543 students.

### 2.3. Variables and Measures

Life satisfaction was assessed using the Cantril ladder [23,24]. Students were asked to use a 10-step ladder to indicate where they would say they are at the moment, with the bottom of the ladder (0) representing the worst way of living, and the top (10) indicating the best possible way of living. Responses were categorised as low life satisfaction (0–5) and high life satisfaction (6–10) [24].

Physical activity was assessed by the number of days youth reported engaging in moderate-vigorous physical activity (MVPA) for at least 60 min. Responses were categorised as at least 60 min/day of MVPA and less than 60 min/day of MVPA, according to WHO recommendations that adolescents spent at least 60 min every day engaging in MVPA [25].

Tobacco use was assessed by the question, “How often do you smoke tobacco?” with four response options “I do not smoke; less than once a week; at least once a week but not every day; and every day”. Answers were dichotomised into tobacco use at least once a week and tobacco use less than once a week.

Frequency of alcohol consumption was assessed by the question, “At present, how often do you drink anything alcoholic, such as beer, wine, spirits, or others?” Youths were given a list of different drinks: beer, wine, spirits, alcopops, or any other drink that contains alcohol. Youths reported the frequency of consumption of every item on the list, and response options ranged from “never” to “every day”. Responses were dichotomised into “alcohol consumption at least once a week”, and “alcohol consumption less than once week”.

Drunkenness was evaluated by asking, “In your lifetime, how often have you had so much alcohol that you became very drunk?” Response options ranged from “never” to “more than 10 times in life”. According to the HBSC international report, responses were categorised as once or fewer, and more than once [26].

The geographic region of residence was derived from the questionnaire and classified into North, Central, and South Italy, according to the Italian National Institute of Statistics (ISTAT) classification.

SES was assessed according to the Family Affluence Scale (FAS), a six-item scale representing a reliable and validated indicator of family wealth (family car ownership, whether adolescents have their own bedroom, number of holidays trips taken in the last year, number of computers owned by the family, dishwasher ownership, and number of bathrooms in the home) [27]. The obtained score (0–13) was categorised as low (0–6), medium (7–9), and high (≥10) SES [26].

### 2.4. Participation in Organised Leisure-Time Activities

Participation in OLTAs was assessed by a yes/no question “In your leisure time, do you do any of the following organized activities?”. Organised activities were detailed as *“…those activities that are done in a sport or another club or organization*”. Possible answers were: (i) individual or team sports associations, (ii) volunteering, (iii) political or youth organisations, (iv) cultural associations (art, musical, scientific associations, etc.), (v) religious groups, and (vi) other. Further explanation within each category were added as per protocol indications.

Answers were categorised in two main groups sport-only and non-sport OLTAs (grouping volunteering, religious, cultural and political associations together). Youths were then classified according to the type of OLTA(s) in which they were involved: not involved in any organised activity (uninvolved), participating in one or more non-sport association (non-sport), participating in one or more sportive association (sport-only), and participating in both sport and non-sport associations (mixed). No-involvement was used as reference category in the analyses.

### 2.5. Data Analyses

Absolute numbers and percentages of sociodemographic (SES) characteristics of the sample and study outcomes (life satisfaction, physical activity, tobacco use, alcohol consumption, and drunkenness) were reported for each of the four OLTA groups, the Chi-square test was performed to evaluate the comparisons among the groups in descriptive analysis. Due to the hierarchical structure of the data, the relationship between OLTA group and the study outcomes (high life satisfaction; at least 60 min/day of MVPA; tobacco use at least 1/week, alcohol consumption at least 1/week, drunkenness at least one in the student’s lifetime) was evaluated by a set of multilevel logistic regressions, adjusted for gender, SES, and geographic area of residence. Results are presented as odds ratios (ORs) and corresponding 95% confidence intervals (CIs). All analyses were carried out by using R version 3.6.0 (Free Software Foundation, Boston, MA, USA), packages *tidyverse*, *janitor*, and *nnet* [28].

## 3. Results

In all the ages (Table 1), girls were most represented in the uninvolved group (23.0%, 30.5% and 39.5% for 11-, 13- and 15-year-olds respectively), while boys were most represented in the mixed group for 11-years old (44.5%) and in the sport-only group in 13- and 15-year-old youth (33.6% and 36.8%). Non-sport OLTAs was among the less represented in all age groups for boys, while the less represented in girls were sport-only and non-sport OLTA for 11-year-olds (19.1% for both), sport-only for the 13-year-olds (20.9%) and mixed OLTA for the 15-year-old girls (18.0%).

Italian youths reported an overall high life satisfaction in all age groups, although the observed significant higher proportion in boys (88.2%, 87.4% and 87.4%) than in girls (87.5%, 79.2% and 77.0%) (*p* < 0.05). Similarly, a higher significant proportion (*p* < 0.001) of boys than girls reported at least 60 min/day of MVPA, with the highest proportion observed in 11-year-old boys (17.2% vs. 10.4% in 11-year-olds; 13.3% vs. 6.7% in 13-year-olds and 10.3% and 6.1% in 15-year-olds). Tobacco use at least once a week increased with increasing age, and was slightly higher in 15-year-old girls (21.5%) than 15-year-old boys (19.6%). Between 11, 13 and 15 years of age, alcohol consumption and drunkenness also increased, and the proportion was significantly higher in boys than girls (alcohol consumption: 6.4% vs. 2.3% in 11-year-olds; 12.4% vs. 6.5% in 13-year-olds and 33.7% vs. 21.1% in 15-year-olds (*p* < 0.001); drunkenness 1.2% vs. 0.4% in 11-year-olds; 4.2% vs. 2.6% in 13-year-olds and 22.6% vs. 18.4% in 15-year-olds (*p* < 0.001)) (Table 1).

Involvement in OLTA, independent of type, showed a positive relationship with reporting high life satisfaction and meeting the recommended level of physical activity, in all age groups and was associated with a reduced likelihood of tobacco consumption in the 13- and 15-year-old groups (Figure 1).

When looking at the relationship between study outcomes and the type of OLTA (Table 2), being in the sport-only and mixed OLTA groups had a positive, significant association with high life satisfaction in all age groups independent of sociodemographic characteristic and geographical region of residence (OR = 1.67 in 11-year-olds, OR = 1.48 in 13-year-olds and OR = 1.55 in 15-year-olds for the sport-only group; OR = 1.95, OR = 1.60 and OR = 1.45, respectively, for the mixed category). Meeting WHO recommendations of at least 60 min/day of MVPA was significantly related to being in the sport-only and mixed groups. This was observed in 11-year-olds (OR = 1.48 for the sport-only group; OR = 1.69 for the mixed group), 13-year-olds (OR = 1.89 for the sport-only group; OR = 1.88 for the mixed group) and 15-year-olds (OR = 2.83 and OR = 3.14, respectively).

Tobacco use at least once a week was inversely associated with OLTA participation with significant results in the 13-year-old group (OR = 0.64 in the sport-only group and OR = 0.72 in the mixed group) and in the 15-year-olds (OR = 0.74 in the non-sport group, OR = 0.72 in the sport-only group, and OR = 0.67 in the mixed group). Participation in OLTAs was associated with alcohol consumption only in the mixed group with a significant slight increase in the odds in 13- and 15-year-olds (OR = 1.26, 95% CI 1.1; 1.5 for 13-year-old and OR = 1.19, 95% CI 1.1; 1.3 for 15-year-old youths). Finally, lower odds of repeated drunkenness were only found among 13-year-olds in the sport-only group (OR = 0.75, 95% CI 0.6; 0.98) and among 15-year-olds in the non-sport group (OR = 0.82, 95% CI 0.7; 0.9). Opposite results were observed for the non-sport group (for the 27) 11-year-olds, with an OR of 1.95 (95% CI 1.18–3.21).

## 4. Discussion

In accordance with other studies, our results revealed that participation in OLTAs was quite a normative experience for Italian youths [10,12]. Among all ages, around 70% and 80% of girls and boys, respectively, were involved in at least one OLTA. The rate decreased with age, but exceeded 60% for both genders.

Our results showed that, overall, participation in OLTAs independent of the type, was positively associated with life satisfaction, with the likelihood of meeting physical activity recommendations and negatively associated with tobacco consumption that, given the low number of current smokers in the youngest, was significant only in 13- and 15-year-olds. Youth engaged in both sport-only and non-sport contexts showed more favourable outcomes across the variables analysed, except for weekly alcohol consumption. In addition, youths who participated in sport-only and mixed OLTAs were less likely to smoke regularly.

Involvement in OLTAs was shown to have a positive impact on social ties and shapes the local sociability domain [3,4], in all considered ages. In so doing, it increases connectedness, offers unique experiences, and promotes identity-related exploration, which ultimately leads to more positive health outcomes [7]. The relationship between involvement in clubs/organisations and life satisfaction has been coherently shown in other studies in which participation was associated with greater self-esteem, fewer health complaints, and greater involvement in cultural, musical, and sports activities [12,29]. It is interesting to note that the association between OLTA and high life satisfaction appears to be present in early ages and it is still present in the oldest, when life satisfaction was shown to decrease, and in particular among girls [12]. To a certain extent, this was explained by people’s need for satisfaction, and especially for relatedness, which has been shown to be fulfilled by OLTA [30,31]. Moreover, participation in organised sports was shown to be favoured with specific self-selection factors as an overall better psychosocial and mental health [32], which, for life satisfaction in the present findings, was found for adolescents involved in the sport-only and mixed groups. Unfortunately given the study characteristics, it is not possible to assess whether factors such as, for example, a pre-existing high life satisfaction, favoured participation and not vice-versa [8].

Furthermore, and consistent with other published studies, our results showed that participation in certain types of OLTA was associated with a slight increase in alcohol consumption, but not in drunkenness [33,34]. Indeed, youths who were participating in sport-only OLTAs were slightly more prone to drink alcohol, supporting the notion that different activities provide adolescents with different developmental experiences or reliance on teamwork. This suggests that some types of OLTA have a contradictory impact on specific health behaviours [11,12]. Participation in structured activities can have many health benefits for adolescents, but such participation has also been linked to negative health behaviours in certain subgroups [13,14], in our case alcohol consumption, constituting the so-called “dark side of social capital” [35].

The relationship between participation in OLTAs and tobacco use appeared to be generally favourable for 13- and 15-year-old youths. Results in the youngest group, because of the low numbers of those reporting to smoke, seem to be less stable. This effect has been partly explained by the fact that adolescents who spend more time under adult supervision are less prone to get involved in less accepted or stigmatised risky behaviours [11]. In line with other studies, these results appear to be more promising when associated with involvement in sportive activities [33].

When it comes to gender differences, in Italy, as in other countries participating in the HBSC study, girls had both a lower occurrence of high life satisfaction, lower levels of physical activity, fewer risky behaviours, particularly those related to alcohol consumption, and a lower level of OLTA involvement [26]. However, it is interesting to underline that, although girls were less involved in OLTAs, all observed positive associations remained statistically significant after adjustment for gender and SES, which highlights the importance of considering the equity of opportunities in a health practice perspective [4] even in the younger ages.

As expected, the significant association observed for reaching the recommended level of physical activity among youth in the sport-only and mixed groups reinforces the indication that meeting physical activity guidelines is easier when youth are offered opportunities to participate in OLTAs supervised by influential adults like, for example, coaches [36].

Moreover, resources that allow people to be actively engaged in their everyday environment have been identified as crucial in meeting physical activity recommendations. In particular including educational intervention to encourage parents to participate with their kids in active transportation and in favouring youths in being engaged in unstructured outdoor activities [37,38].

Given that the highest drop in physical activity tends to happen between 13 and 15 years of age, particularly in girls, [26,39] the WHO Global Action Plan on Physical Activity stresses that public health sectors must adopt a multisectoral, integrated approach to re-engage youths within and beyond this age group. This WHO plan aims to reduce physical inactivity by promoting a wide range of comprehensive opportunities, such as fair and safe access to places and spaces, easy access to clubs and outdoor sports facilities [40]. A previous study reported that clubs with higher proportions of private funds emphasised competition and achieving top sports results, whereas clubs that focus on the local population are often forced to rely on volunteers [41]. Available evidence further suggests that public investment in private clubs is needed and could motivate these clubs to reaffirm their commitment to engaging and serving the community as a whole, instead of focusing only on a results-oriented economic perspective [42]. Given that OLTA in Italy are mainly offered through the efforts of the third-sector and largely dependent on household financial contribution [21], it would require a higher financial investment from the public sector, shifting, for example, available funds from competitive sports to recreational/educational OLTAs [42,43]. This may help achieve a broader aim to reduce inequities in social and environmental opportunities. Relying upon higher public financial compensation, community clubs that organise OLTAs could have a favourable impact on health behaviours among youths, not just in terms of reaching recommended physical activity levels or fighting substance use, but also in terms of favouring health practices and life satisfaction in developmental ages [10,33,43,44].

### Study Strength and Limitations

The results of this study should be read in light of its limitations and strengths; first and foremost, those related to the study design. The HBSC study is a cross-sectional study and, therefore, it does not allow us to draw any conclusions about causation. Furthermore, the dichotomization/categorization of some of the variables under study (life satisfaction, OLTA involvement, and so on) has inevitably led to a loss of information. Despite that, we decided to follow faithfully the HBSC protocol recommendations [22] in order to allow comparability with other studies among the other HBSC participating countries. This choice could have introduced further specific limits, in particular on how OLTAs were treated. Participation in OLTAs was treated by dichotomizing the youth declaration of involvement, without considering other important factors related to participation such as breadth, intensity and duration [8,18]. Moreover, as involvement was shown to be related to pre-existing differences among the involved and the non-involved groups (as demographic, contextual and individual factors), treating the non-involved as a single homogeneous group can produce misleading results on the overall effect of involvement [8,45]. Finally, our results are limited to Italy (only one of the nearly 50 countries participating in the HBSC study)

Among the main strengths, it must be evidenced that the study is based on a robust survey, the HBSC study, and that the 2014 wave of this study in Italy represents the largest (approximately 10-fold the other countries) sample size available in these developmental ages [46]. As such, it can positively contribute to the expanding research in this field.

## 5. Conclusions

Results showed that adult-supervised OLTAs represent a practical and affordable opportunity for fair and safe access to places and spaces that can shape community health practices in youths. Participation in OLTAs was positively associated with several important positive health outcomes such as perceived life satisfaction, compliance with international physical activity recommendations, and reduced weekly tobacco use. Being engaged in more than one OLTA, in particular any kind of sportive OLTA, showed positive effect on all of the outcomes under study, except for weekly alcohol use.

Higher participation in OLTAs, according to other recent studies, constitutes a value in itself, as it enriches youths’ lives by increasing connectedness, offering unique experiences, and promoting identity-related exploration, which could ultimately lead to more positive health outcomes. Our results suggest that increasing the opportunity to engage youth in out-of-school OLTAs can affect positive health practices. In countries where resources for physical activities are left to the private/non-profit sector initiatives, as in Italy, promoting the shift of available public funds to recreational/educational OLTAs, rather than favouring those with higher competitive sports results, has the potential to reduce inequities in the access to social and environmental opportunities in youth.

## Figures and Tables

**Figure 1 ijerph-17-06637-f001:**
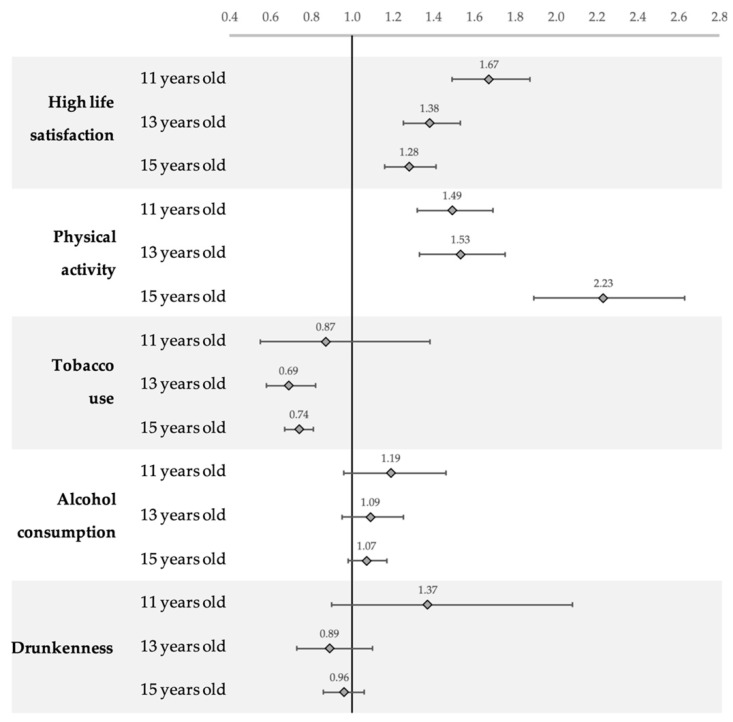
OR and 95% confidence intervals by age group for involvement in organised leisure time activities (OLTA) and study outcomes ^§^. Health Behaviours in School-aged Children (HBSC), Italy 2014. All analyses were mutually adjusted for gender, sociodemographic characteristics and geographic area of residence; youth’ school class-level was used in all models. ^§^—high life satisfaction (≧6), physical activity (1h/day), tobacco use at least once a week, alcohol consumption at least once a week, and drunkenness (more than once in life) in 11-, 13- and 15-year-old youth.

**Table 1 ijerph-17-06637-t001:** Sociodemographic characteristics, involvement in different types of organised leisure-time activities (OLTA) and study outcomes. Health Behaviours in School-aged Children (HBSC), Italy 2014.

	11 Years Old		13 Years Old		15 Years Old	
	15,794		15,019		12,730	
	Boys 8017	Girls 7777	Boys 7396	Girls 7623	Boys 6282	Girls 6448
	*n* (%)	*n* (%)	*n* (%)	*n* (%)	*n* (%)	*n* (%)
Residence: North of Italy	4259 (53.1)	4076 (52.4)	3795 (51.3)	3861 (50.6)	2972 (47.3)	3089 (47.9)
Residence: Central Italy	1440 (18.0)	1470 (18.9)	1413 (19.1)	1457 (19.1)	1273 (20.3)	1334 (20.7)
Residence: South of Italy	2318 (28.9)	2231 (28.9)	2188 (29.6)	2305 (30.2)	2037 (32.4)	2025 (31.4)
FAS: Low ^‡^	1614 (20.7)	1728 (22.7)	1252 (17.2)	1545 (20.5)	1099 (17.9)	1243 (19.5)
FAS: Medium ^‡^	4040 (51.9)	3984 (52.3)	3779 (51.9)	3956 (52.5)	3214 (52.4)	3397 (53.4)
FAS: High ^‡^	2134 (27.4)	1900 (25)	2243 (30.8)	2032 (27.0)	1825 (29.7)	1717 (27.0)
Type of OLTA: Uninvolved	1360 (17.4)	1734 (23.0)	1494 (20.6)	2268 (30.5)	1553 (25.1)	2493 (39.5)
Type of OLTA: Non-sport	917 (11.7)	1440 (19.1)	923 (12.7)	1564 (21.0)	865 (14.0)	1387 (22.0)
Type of OLTA: Sport-only	2065 (26.4)	1440 (19.1)	2436 (33.6)	1555 (20.9)	2278 (36.8)	1294 (20.5)
Type of OLTA: Mixed	3478 (44.5)	2941 (38.9)	2401 (33.1)	2044 (27.5)	1499 (24.2)	1136 (18.0)
High life satisfaction	7067 (88.2)	6806 (87.5)	5455 (87.4)	6009 (79.2)	5455 (87.4)	4930 (77.0)
At least 60 min/day of MVPA	1377 (17.2)	811 (10.4)	977 (13.3)	505 (6.7)	644 (10.3)	388 (6.1)
Tobacco use at least 1/wk	78 (1.0)	29 (0.4)	367 (5.0)	368 (4.8)	1232 (19.6)	1389 (21.5)
Alcohol consumption at least 1/wk	511 (6.4)	175 (2.3)	920 (12.4)	499 (6.5)	2116 (33.7)	1358 (21.1)
Drunkenness at least once in life	95 (1.2)	33 (0.4)	312 (4.2)	201 (2.6)	1423 (22.6)	1186 (18.4)

^‡^ Family Affluence Scale. Missing in FAS (11-year-olds *n* = 349; 13-year-olds *n* = 212; 15-year-olds *n* = 235); in OLTA (11-year-olds *n* = 419; 13-year-olds *n* = 334; 15-year-olds *n* = 225); in life satisfaction (11-year-olds *n* = 129; 13-year-olds *n* = 102; 15- year-olds *n* = 81); in physical activity (11-year-olds *n* = 262; 13- year-olds *n* = 145; 15- year-olds *n* = 94); tobacco use (11-year-olds *n* = 26; 13- year-olds *n* = 31; 15- year-olds *n* = 45); alcohol consumption (11-year-olds *n* = 43; 13- year-olds *n* = 16; 15- year-olds *n* = 21); drunkenness (11-year-olds *n* = 69; 13- year-olds *n* = 65; 15- year-olds *n* = 37).

**Table 2 ijerph-17-06637-t002:** Multilevel logistic regression analyses of involvement in organised leisure time activities (OLTA) and study outcomes ^§^. Health Behaviours in School-aged Children (HBSC), Italy 2014.

	High Life Satisfaction		Physical Activity		Tobacco Use		Alcohol Consumption		Drunkenness	
	*n* (%)	OR (95% CI)	*n* (%)	OR (95% CI)	*n* (%)	OR (95% CI)	*n* (%)	OR (95% CI)	*n* (%)	OR (95% CI)
**11 years old**										
Uninvolved	2569 (18.5)	1	286 (13.1)	1	21 (19.6)	1	111 (16.2)	1	22 (17.2)	1
Non-sport	2026 (14.6)	1.21 (1.03–1.41)	238 (10.9)	1.07 (0.9–1.27)	14 (13.1)	0.78 (0.41–1.52)	98 (14.3)	1.24 (0.94–1.63)	27 (21.1)	1.95 (1.18–3.21)
Sport-only	3129 (22.6)	1.67 (1.44–1.94)	521 (23.8)	1.48 (1.27–1.71)	20 (18.7)	0.84 (0.47–1.48)	141 (20.6)	1.05 (0.81–1.35)	15 (11.7)	0.83 (0.49–1.41)
Mixed	5823 (42.0)	1.95 (1.71–2.23)	1076 (49.2)	1.69 (1.48–1.93)	45 (42.1)	0.89 (0.54–1.46)	318 (46.4)	1.24 (0.99–1.55)	58 (45.3)	1.39 (0.89–2.15)
**13 years old**										
Uninvolved	2983 (24.2)	1	236 (16.2)	1	228 (31.9)	1	317 (22.7)	1	127 (26.3)	1
Non-sport	1977 (16.1)	1.01 (0.90–1.16)	141 (9.7)	0.90 (0.73–1.12)	130 (18.2)	0.83 (0.67–1.04)	226 (16.2)	1.07 (0.89–1.27)	73 (15.1)	0.85 (0.63–1.14)
Sport-only	3462 (28.1)	1.48 (1.30–1.68)	518 (35.6)	1.89 (1.61–2.23)	157 (22.0)	0.64 (0.51–0.79)	348 (24.9)	0.92 (0.78–1.08)	112 (23.2)	0.75 (0.58–0.98)
Mixed	3883 (31.6)	1.60 (1.41–1.81)	559 (38.4)	1.88 (1.60–2.21)	200 (28.0)	0.72 (0.59–0.88)	506 (36.2)	1.26 (1.08–1.47)	171 (35.4)	1.05 (0.82–1.33)
**15 years old**										
Uninvolved	3147 (30.8)	1	161 (15.8)	1	995 (38.6)	1	1003 (29.5)	1	810 (31.8)	1
Non-sport	1731 (16.9)	0.93 (0.83–1.06)	108 (10.6)	1.20 (0.93–1.53)	442 (17.1)	0.74 (0.65–0.84)	560 (16.5)	0.98 (0.87–1.11)	393 (15.4)	0.82 (0.72–0.94)
Sport-only	3093 (30.3)	1.55 (1.36–1.75)	416 (40.9)	2.83 (2.33–3.43)	670 (26.0)	0.72 (0.64–0.81)	1011 (29.7)	1.00 (0.90–1.12)	820 (32.2)	1.07 (0.95–1.19)
Mixed	2246 (22.0)	1.45 (1.27–1.67)	331 (32.6)	3.14 (2.57–3.82)	472 (18.3)	0.67(0.59–0.76)	828 (24.3)	1.19 (1.06–1.33)	525 (20.6)	0.88 (0.77–1.00)

Uninvolved (no-OLTA) category was used as reference. Statistically significant results are in bold; All Analyses were mutually adjusted for gender, sociodemographic characteristics and geographic area of residence; youth’ school class-level was used in all models. ^§^—high life satisfaction (≥6), physical activity (1 h/day), tobacco use at least once a week, alcohol consumption at least once a week, and drunkenness (more than once in life) in 11-, 13- and 15-year-old youths.

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
