# Peer review of "The Contribution of Organised Leisure-Time Activities in Shaping Positive Community Health Practices among 13- and 15-Year-Old Adolescents: Results from the Health Behaviours in School-Aged Children Study in Italy"

_ijerph, 2020, doi:10.3390/ijerph17186637_

Round 1

Reviewer 1 Report

The authors present an interesting cross-sectional study that benefits from a nationally representative sample of young people and uses validated questionnaire items. However, it is unclear what the unique focus is of this study compared to previous published literature on the relationships between OLTA participation and associated outcomes. Have these outcomes been investigated in combination before? Nevertheless, the fact that these relationships are reported for a country less represented in the existing literature is informative. To this end, the manuscript could be strengthened by further contextualising the outcome variables in Italy, and how they present in comparison to other countries. Similar to this point, additional contextual information on how OLTA participation is normally accessed (through schools or outside schools for example) in Italy provides the international audience with some understanding of how the results can be applied and evaluated; particularly from a public health perspective. 

Other general points include: 

  • I suggest the authors change the description of the ‘sportive’ category to ‘sport only’ and ‘non-sport’ through out 
  • There are formatting errors, grammatical errors and inconsistent referencing used. 
  • Overall, the tone of language suggesting causality could be ‘toned down’. This is a cross-sectional study and effects of any direction cannot be inferred. Additionally, self-selection factors cannot be accounted for and should be discussed as a limitation. 

Further specific points are outlined below according to section: 

Abstract: 

  • Typo - line 19 
  • It is not clear how participation in OLTAs was conceptualised and measured 
  • Interpretation of the findings would be supported by clearly stating how participants were compared on outcomes according to OLTA participation type (I.e. sport, non-sport, mixed and no OLTA) 

Introduction 

  • The opening sentence is difficult to read and could be more specific. Consider combing this sentence with information from the second sentence. 
  • Please provide a reference to following statement “A recent analysis showed that unhealthy behaviours among adolescents tend to reflect collective lifestyles, rather than an individual’s choices” (line 43) 
  • I am unclear about the term “socially possible experiences” and what this encompasses. Please rephrase/revise and provide examples so your rationale and arguments are clearer. 
  • The term ‘broader community perspective’ (line 50), seems to be synonymous with an ecological approach. Is this what the authors mean or is this meaning different? 
  • Line 58: Please expand more on this statement by providing detail of the study and a reference: “When studied, bonding and bridging ties were both found to be important in shaping health practices”.  
  • In what ways are these relationships different depending on gender and socioeconomic status (line 79-80)? Please expand on the nature of these relationships. 
  • I was surprised  that more literature wasn’t discussed in regards to life satisfaction. This is a dependent variable of focus but the introduction discusses primarily relationships between OLTA and health behavioursHow might forms of OLTA activity associate differently with life satisfaction and why? 

Methods 

  • As mentioned in the discussion, OLTA participation changes with age. The authors have valuable information in being able to disentangle these relationships with having a younger cohort. And although health behaviours might be limited at a young age (which may be due to measuring regular use of for tobacco and alcohol, rather than ever use), there still may be useful inferences to make about OLTA participation and life satisfaction with the younger age group. 
  • Was the high and low derived variable for life satisfaction based on a median split (line117)? Please provide more information on why this variable was changed into a dichotomous variable. 
  • Please provide all the response options for tobacco use (122), it is not necessarily clear what the other options were. 
  • Data analysis: how was OLTA participation entered into the logistic regressions? Were dummy variables created for each response category and a reference category used? 

Results 

  • Line 152 – unless authors have a strong rationale, I suggest renaming ‘socioeconomic’ to ‘sociodemographic’ characteristics 
  • The text from Table 1 (line 163-166) describes patterns by identifying just the highest percentages for different groups. This only skims the surface of what the data is showing, as low percentages are also noteworthyFor more clarity, I suggest discussing how OLTA participation presents for males and females (high percentages and low percentages irrespective of age), and then what differences in patterns emerge (or don’t emerge) across the two age groups. 
  • Tables titles are too long. I suggest shortening for clarity. 
  • Table 2: I found the ‘N’ being reported in the table confusing. What does this represent? 
  • Please refrain from using first person (line 183) 
  • Line 192: please use ‘association’ when explaining relationships and not ‘effect’ as this is a cross-sectional study and causality cannot be inferred. 
  • I am unclear how to interpret the table given the lack of information on how OLTA participation was entered into the logistic regression. If ‘uninvolved’ is the reference group in the model, than the OR’s provide the likelihood of other groups doing that particular outcome more or less than the reference group. If this is the case, it would therefore be incorrect to infer that participation in any OLTA is positively associated with tobacco use (line 199-200). In addition, the statement “Participation in OLTAs was not associated with alcohol consumption” (line 203) can not hold true if those who participated in mixed OLTA were more likely to engage in alcohol use compared to those who did not engage in any OLTA.  

Discussion 

  • Please revise the statement that OLTA participation showed an ‘impact’ (line 213) on outcomes. As this is a cross-sectional study, causality cannot be inferred. 
  • The statements made (line 212-215) relate to general benefits of participation in OLTAs, however the analyses compared no OLTA participation to every other group. It therefore cannot be said that those who participated in OLTAs were less likely to get drunk, as this relationship was only apparent when comparing no OLTA participation and non-sport participation. For the other OLTA participation types, there was no difference in risk when comparisons are made to the no OLTA participation group. If this study wanted to test whether participation generally (irrespective of type) was associated differently with outcomes, then a single dichotomous variable indicating OLTA participation vs no OLTA participation could be used instead. I suggest these general statements be deleted, with the key message of the study being the statement on line 216. 
  • Much of the discussion surrounding life satisfaction would be more useful if situated in the introduction to support a clearer argument and rationale for this focus.  
  • Can the authors please elaborate on this statement and provide a reference (line 257) including helping active relatives engage youths in unstructured outdoor activities”.   
  • Can the authors please elaborate on the limitations of their study in terms of the cross-sectional design and influence of self-selection factorsFor example, individuals with higher life-satisfaction may be more likely to engage with OLTAs in the first place, or have higher FAS scores which may mean more disposable income to encourage engagement in OLTAs from a young age. For a more detailed discussion on the role of self-selection factors generally please see: 
  •  Bohnert, J. Fredricks, E. Randall (2010) Capturing unique dimensions of youth organized activity involvement. Review of Educational Research, 80, pp. 576-610 
  • Fredricks, J. A., & Eccles, J. S. (2006). Is extracurricular participation associated with beneficial outcomes? Concurrent and longitudinal relations. Developmental psychology, 42(4), 698.).  
  • Some components of the conclusion go beyond the data and what the study investigated. While I agree with the authors that forms of OLTA participation provide developmental opportunities for young people, and can potentially address widening inequalities. However, the authors did not assess whether OLTAs are practical or affordable. A stronger argument for the re-allocation of resources could be made if the analyses were more focused on the role of FAS and how this shaped associations. 

Author Response

Please see attachment below with all point-by-point detailed response.

Reviewer 2 Report

Review Report submitted to IJERPH

Manuscript ID: ijerph-861769

Title: The contribution of organised leisure-time activities in shaping positive community health practices among 13- and 15-year-old adolescents: results from the Health Behaviours in School-aged Children study in Italy

Comments to authors

General

Using data from a representative sample from the Italian 2013/2014 Health Behaviours in School-aged Children study, the authors showed that participation in organised leisure-time activities was associated with favourable impact on some health behaviours of 13- and 15-year-old students.

I observed that there was no spacing before the in-text references.

I hereby submit my comments that could improve the study. 

Abstract

Methods: Data were collected according to the HBSC study protocol. Hierarchical logistic regression models were used. What for? Be specific 

Results:

“Furthermore, youths participating in OLTAs were more likely to meet physical activity recommendations and report lower rates of tobacco use and drunkenness.” I suggest you report the ORs and corresponding 95% CI for these variables. Provide the reader with adequate information.  

Conclusion: “Coherently with other studies”. Delete this statement.

“Available evidence further suggests that public investment in sport clubs could motivate them to reaffirm their commitment to engaging and serving the community as a whole and further reducing inequities in social and environmental opportunities.” Although you may be right in, your conclusion should be based on the available data from the present study, not what is already known.

Introduction

Page 2, lines 43-44: missing references

Page 2, lines 70-71: missing in-text references

Data and Methods

Page 3, line 92: which research committee?

Page 3, lines 107-108: “The dataset was cleaned to ensure consistency in the variables under analysis, and 4,265 students with incomplete questionnaires.” The sentence needs editing.

Page 3, line 94: delete one “Informed consent”

Page 3, line 123-125: “The four response options ranged from “I do not smoke” to “every day”.  Answers were dichotomised into tobacco use at least once a week and tobacco use less than once a week.” What about the "I do not smoke" option? And “never” for alcohol consumption? It is not clear how these responses were handled in the analysis? Were they excluded? The section needs editing.  

Data analyses section needs more information. For example, did you test for data distribution?  

Results

Page 6, lines 183-190: The authors did not specify whether the observed differences were significant. It would be helpful to provide such information.  

Page 6, line 199 & 202: This was contradictory. Positive associations indicate that being involved in one increases the odds of the other. However your statement "this participation reduced the odds of tobacco use in both age groups (13-year- olds: OR=0.64 in the sportive group and OR=0.72 in the mixed group; 15-year-olds: OR=0.74 in the non-sportive group, OR=0.72 in the sportive group, and OR=0.67 in the mixed group)" suggests an inverse association. Please clarify.

Page 6, lines 202-204: “Participation in OLTAs was not associated with alcohol consumption; only the mixed group was linked to a slight increase in risk in both age groups…..” From the results, participation in OLTAs was associated with alcohol consumption but only in the mixed group. This should be clearly stated.

Discussion 

Page lines 228-230: “Moreover, participation in organised sports was shown to be predictive of positive mental health in adolescence(Vella, Swann, Allen, Schweickle, & Magee, 2017), which matches the present findings, as higher life satisfaction was found only in adolescents in the sportive and mixed groups.” Could the authors explain this? Are they associating positive mental health to higher life satisfaction? Positive mental health may lead to higher life satisfaction. But the present study did not assess mental health. The implication is hence misleading. 

Page 7, lines 274: “According to our and other study results, relying upon higher public financial compensation,……” The study did not collect data on funding sources for the OLTAs so why the reference to “our study”?

Study limitations

Page lines 280-281: “The results of this study should be read in the lights of its limitations and strengths first and foremost, those related to the study design”…….. Rephrase. The section also had strengths. The authors should re-consider the sub-heading.

Conclusion

Page 8, lines 302-306: “Our results suggest that, in countries where resources for physical activities are left to the private/non-profit sector, promoting the shift of available public funds from competitive sports to recreational/educational OLTAs has the potential to reduce inequities in the access to social and environmental opportunities in youth.” This is not supported by the present study.

Author Response

(The authors gave the same response as above.)

Reviewer 3 Report

Dear authors, it is a pleasure for me to review such a well-elaborated, planned, and developed work.

However, before giving final approval certain aspects should be considered by the authors for the improvement of this manuscript.

Firstly, although the introduction is well structured, it is necessary to replace some of the references used with more current ones, as well as to refer to the latest studies carried out on adolescent populations in several European countries.

Concerning the methodology, I have no appreciation to make.

Based on the results, I recommend the authors to divide table 1 and 2 into two smaller tables to facilitate the reading and understanding of the table for the readers. In the same way, certain data could be understood to a greater extent by the readers if they were represented using figures.

As for the discussion and conclusions of the article, I suggest improving and extending them based on the results obtained. Without a doubt, more information can be extracted from the different variables analyzed.

Finally, I recommend the authors to review the bibliographic references used and to adapt them based on the regulations of the journal.

Author Response

(The authors gave the same response as above.)

Round 2

Reviewer 3 Report

Once you have followed the first and second reviewer´s comments, it is a pleasure for my give my approval to the publication of the manuscript.